# pH Sensitive Erythrocyte-Derived Membrane for Acute Systemic Retention and Increased Infectivity of Coated Oncolytic Vaccinia Virus

**DOI:** 10.3390/pharmaceutics14091810

**Published:** 2022-08-28

**Authors:** Kaelan T. Samoranos, Alexandra L. Krisiewicz, Bianca C. Karpinecz, Philip A. Glover, Trevor V. Gale, Carla Chehadeh, Sheikh Ashshan, Richard Koya, Eddie Y. Chung, Han L. Lim

**Affiliations:** 1Coastar Therapeutics Inc., San Diego, CA 92121, USA; 2Neon Insights Consulting, Auburndale, MA 02466, USA; 3Department of Obstetrics and Gynecology, Pritzker School of Medicine, The University of Chicago, Chicago, IL 60637, USA

**Keywords:** cancer, pH responsive, cell membrane, oncolytic virus, vaccinia, erythrocyte, drug delivery, nanotechnology, systemic administration, smart materials

## Abstract

Oncolytic viruses have emerged as a promising modality in cancer treatment given their high synergy with highly efficient immune checkpoint inhibitors. However, their potency is limited by their rapid in vivo clearance. To overcome this, we coated oncolytic vaccinia viruses (oVV) with erythrocyte-derived membranes (EDMs), hypothesizing that they would not only remain in systemic circulation for longer as erythrocytes would when administered intravenously, but also respond to environmental pH cues due to their membrane surface sialic acid residues. For this, we developed a model based on DLVO theory to show that the acidic moieties on the surface of EDM confers it the ability to respond to pH-based stimuli. We corroborate our modeling results through in vitro cell culture models and show that EDM-coated oVV infects cancer cells faster under acidic conditions akin to the tumor microenvironment. When EDM-coated oVVs were intravenously injected into wild-type mice, they exhibited prolonged circulation at higher concentrations when compared to the unprocessed oVV. Furthermore, when EDM-coated oVV was directly injected into xenografted tumors, we observed that they were suppressed earlier than the tumors that received regular oVV, suggesting that the EDM coating does not hinder oVV infectivity. Overall, we found that EDM was able to serve as a multi-functional encapsulant that allowed the payload to remain in circulation at higher concentrations when administered intravenously while simultaneously exhibiting pH-responsive properties.

## 1. Introduction

In the past two decades of clinical testing, multiple attempts to develop viruses as oncolytic agents have yielded mixed results; while some oncolytic viruses (OVs) have made it to the market, there have also been notable failures in other developments [1,2]. Today, OVs continue to garner interest because of their mechanisms of action; the multimodality of OV therapy includes the lytic infection activity that exposes tumor neoantigens while simultaneously delivering transgenes for tumor-specific expression to enhance host anti-tumor immune response [3,4]. On top of that, recent studies have shown that OVs demonstrated remarkable synergy with the efficacious immune checkpoint inhibitors that has led some to believe that this combination therapy may become a new paradigm in treating cancers [5,6,7]. Of the many OVs that have been studied, oncolytic vaccinia viruses (oVVs) have emerged as a popular choice for their ability to quickly induce oncolysis in cancer cells, and its potential to carry multiple transgenes in its viral DNA, and its short life cycle that takes place entirely in the cytoplasm lowers the risk of genomic integration [8].

However, some obstacles remain; oVVs, being obligate intracellular parasites, are readily and rapidly cleared by a variety of host defenses [9]. This phenomenon represents a key challenge that is especially difficult for intravenous (IV) administration; the IV administration of OVs would be highly desirable to allow these viruses to also target metastatic secondary tumors. Additionally, while oVVs are often thought of as enveloped viruses, their manufacturing process in larger scales often leads to a product that contains a mixed population of enveloped or unenveloped oVVs of which the exact proportion of each in the mix is typically not characterized. To add to that, while both enveloped and unenveloped oVV exhibit infectiousness in cancer cells, they have different physicochemical characteristics and biomolecular signatures that lend to their rapid clearance in vivo.

To address this issue, multiple OV coating/stealthing strategies have been tested including cellular, polymer, and nanoparticle modifications. Cellular stealthing has included variously sourced stem cells to deliver OVs systemically to tumors, often taking advantage of the inflammation-homing activity of mesenchymal stem cells and has been tested with adenovirus [10,11,12], herpes virus [13,14], and vaccinia virus [15]. Similarly, polymers such as polyethylene glycol or polyethyleneimine were evaluated for their ability to shield OVs including adenovirus [16], measles virus [17], vesicular stomatitis virus [18], and vaccinia virus [19]. Finally, nanotechnology inspired viral modifications have also been tested for the ability to protect OVs including adenovirus [16,20]. While some benefit is gained, the widespread adoption of these modifications has not been realized, as it is still unsure whether any of the modifications may themselves elicit immune responses that are not always immediately clear to the medical community [21,22,23].

More recently, studies have shown that cell membrane-based encapsulations of adenoviruses can confer them a pseudo-envelope, even though adenoviruses are not naturally enveloped viruses. This pseudo-enveloped virus has exhibited meaningful benefits, such as prolonged circulating half-life, immune evasion, and tumor specificity. Here, we report that this can also be applied onto the conventionally enveloped oVV by coating the oVV with erythrocyte-derived membrane (EDM). By using the nonimmunogenic erythrocyte membrane as the oVV encapsulant, we can take advantage of the privileges conferred by its receptor expression, many of which allow the erythrocyte to remain in circulation for up to 3–4 months [24,25]. Additionally, apart from the improved circulatory retention, we observed pH responsiveness in the EDM-coated oVVs, which showed potential for the targeted and improved infectivity of the virus into cancer cells within the acidic tumor microenvironment.

## 2. Materials and Methods

### 2.1. The Cells, Viruses, and Reagents

The human small cell adenocarcinoma line A549, human breast adenocarcinoma line MCF-7, and African green monkey kidney cell line VERO E6 were obtained from the American Type Culture Collection (Manassas, VA, USA). We maintained these cell lines in Dulbecco modified Eagle’s media (DMEM) containing L-glutamine (Gibco, Waltham, MA, USA) amended with 10% fetal bovine serum (Gibco, Waltham, MA, USA) and 1X antibiotic–antimycotic (Gibco) in a humidified cell culture incubator at 37 °C and 5% CO_2_. oVV derived from the Western Reserve strain with preferential tumor replication by the disruption of the thymidine kinase and vaccinia growth factor genes, with an inserted gene encoding an enhanced green fluorescent protein (eGFP) was kindly provided by Dr. Richard Koya (University of Chicago) and its construction is described in detail elsewhere [26]. In brief, the constructed oVV-eGFP virus was then amplified on HeLa cells, lysed to collect the product, purified over the sucrose gradient to remove cellular debris, and titered. The expression of eGFP in oVV-eGFP–infected 4T1 cells was confirmed by immunofluorescence microscopy. The oVV-eGFP manufactured this way is hereafter referred to in this manuscript as “oVV”.

### 2.2. Derivation of EDM from Human/Mouse Whole Blood

Studies conducted in vitro and in the xenograft model were performed with erythrocyte-derived membrane (EDM) derived from human donor erythrocytes, while studies conducted in vivo with wild-type mice were performed with EDM derived from mouse erythrocytes. Both EDM materials were generated as previously described for the enrichment of cellular membrane by others [27]. Briefly, for human erythrocytes, we procured donor Type O Rh-negative whole human blood from BioIVT (Westbury, NY, USA). The red blood cells were separated, osmotically lysed, and the membranous fraction was recovered by tangential flow filtration. In contrast, for mouse erythrocytes, the recovery of the membranous fraction was accomplished using centrifugation at 15,000 rcf. We characterized the recovered membrane for the total protein content by BCA assay following the manufacturers’ protocol (Pierce BCA Protein Assay, Thermo) and stored frozen at or below −20 °C until use.

### 2.3. Encapsulation Procedure

Encapsulation of oVV was achieved by fine membrane extrusion as has been previously described by others [28,29]. Briefly, we sanitized an extrusion apparatus consisting of two glass syringes, a membrane support housing (Avanti, Alabaster, AL, USA), and 1.0 µm-pore polycarbonate membrane (Whatman, Maidstone, UK) by soaking each component in 70% isopropanol followed by a rinse with sterile DI water (Figure 1A). The pre-extrusion volume contains oVV obtained from the manufacturers (“oVV”) in EDM at approximately a 1:100 mass ratio based on the whole protein content. This mix of EDM plus oVV was then processed by repeatedly passing back and forth through the membrane, propelled by syringe pumps. The process is illustrated in the schematics shown in Figure 1B and Figure 1C. Coated oVV (“EDM-coated oVV”) were deposited in sterile 1.5 mL tubes and used immediately. We use qPCR, as detailed in the protocol below, to quantify the number of copies of oVV DNA and EDM-coated oVV DNA, to ensure that no oVV is lost during the encapsulation process.

### 2.4. ELISA

Enzyme-linked immunosorbent assay (ELISA) detecting oVV was performed using the protocol that follows. We procured Rabbit anti-vaccinia HRP-conjugated polyclonal antibody as a detector (Abcam#87387), which was generated in rabbits against whole vaccinia virus. Serving both as a positive control and as a standard curve, freshly thawed oVV is first diluted to the same effective concentration as the extruded sample used for infecting cells; which is 3.00 × 10^6^ pfu/mL. From this, we further diluted the sample to yield concentrations of: 1.50 × 10^6^ pfu/mL, 3.00 × 10^5^ pfu/mL, and 3.00 × 10^4^ pfu/mL. A negative control comprising only EDM without any oVV was used and labelled as 0 pfu/mL. These samples are then used for coating an ELISA plate (*n* = 3). After incubation, we decanted all samples from the plate and washed 4 times with PBS. We blocked the plate with a solution of PBS plus 5% BSA (0.05 g/mL) and allowed the plate to incubate for 1 hour at room temperature. The block was decanted, and the plates were washed 4 times with PBS. We added polyclonal anti-Vaccinia HRP-conjugated antibody diluted to 4 µg/mL in blocking buffer to the plate and again incubated the plate for 1 hour at room temperature. After, the antibody was decanted. The plate was again washed 4 times with PBS. Finally, 100 µL of TMB solution (BioLegend, San Diego, CA, USA) was added to each well. We monitored each assay well for development and stopped the TMB reaction with 1.0 M hydrochloric acid. Absorbances of each well were read with a BioTek plate reader at 450 nm.

This ELISA is also used to determine the coating efficiency using the following steps that are performed alongside the concentration curve protocol. The extrusion product, which we conservatively assumed contains a mix of EDM-coated oVV and untouched oVV, is directly used to coat the bottom of other wells on the same ELISA plate, and then subjected to the same protocol. We hypothesize that of the mixture, only oVV will contain any reactivity against the antibody, and produce a positive HRP signal, while the EDM-coated oVV will not.

### 2.5. DLS/Zeta Potential

Dynamic light scattering (DLS) particle sizing and zeta potential measurements were both measured using a Malvern Panalytical Zetasizer Nano ZS in accordance with the manufacturer’s user manual. For particle size measurements, we gently pipette-mixed the sample of interest and aliquoted 300 µL of the sample into a disposable semi-micro-Vis cuvette (EP0030079353). We measured the particle size using the optimized particle size protocol from the Zetasizer program. To measure the zeta potential, we gently pipette-mixed the sample of interest, diluted the sample 1:10 in DI water, aliquoted 800 uL of the diluted sample into a folded capillary cuvette (DTS1070), and finally measured the zeta potential using the optimized particle size protocol from the Zetasizer program. This was repeated across samples at different pHs. We adjusted the pH of the sample using hydrochloric acid. We again repeated the measurements at different concentrations of EDM and allowed the sample to incubate at room temperature for different amounts of time after dilution to ensure that the samples reached equilibrium.

### 2.6. Calculating Relative Interparticle Forces at Different pHs

In order to determine the relative interparticle forces at different pHs, we first obtained a relative interparticle distance. Others previously devised a method to approximate the interparticle distance in colloidal mixtures utilizing particle concentration (n), which can be derived from the average relative particle volume (V_avg_) [30]. We determined the EDM’s interparticle distance using the following:d_z-avg_^3^ ∝ V_avg_,(1)
where d_z-avg_ is the average diameter of the particles. We obtained d_z-avg_ using the DLS measurements, and that diameter scales cubically with volume. Then, we obtained the number concentration of particles in the system (n) by dividing the mass concentration of particles (C) by the relative average particle volume (V_avg_):n ∝ C/V_avg_,(2)
where n represents the number concentration. Using this, as others have done, we then very coarsely estimated the relative interparticle distance (a) as the cube root of the number concentration of particles (n),
a ∝ 1/n^1/3^.(3)

We assume that the zeta-potential of the suspension (ζ) is proportional to the relative magnitude of charge of the particles’ surface (q_i_), given by:ζ ∝ q_i_.(4)

Together, we approximate the relative repulsive force using the scalar form of Coulombic repulsion, given by:F = q_1_q_2_/a^2^,(5)
where F is the relative repulsive force between particles [30]. As every instance of q_i_ originates from the same species of particle, for the purpose of estimation, we reduce the equation to
F = q^2^/a^2^.(6)

Using this framework for analysis, we determine the relative interparticle distance and its corresponding relative interparticle force for every measured d_z-avg_ and ζ quantity obtained across its various pHs and equilibrium times.

### 2.7. pH-Mediated Infectivity Assay

For in vitro studies, A549, Vero E6, and MCF-7 cells were plated into 24-well plates at a density of 100,000 cells per well and incubated overnight for adhesion. The day following plating, we used DMEM (sans serum) “as is” for the neutral pH condition. For the acidified condition, to best mimic the pH conditions of a tumor microenvironment, we acidified DMEM with hydrochloric acid to a pH of ~5.2. Subsequently, we reconstituted the bare and coated viruses into their respective conditions, neutral or acidified, to an equivalent MOI of 0.5 and added to the cells for our infection phase, comprising 1 hour of incubation in a humidified incubator at 37 °C, 5% CO_2_. Following the infection phase, we aspirated the virus preps and washed cells with growth media 3 times to rinse residual virus preps and prevent further infection. For A549, Vero E6, and MCF-7 cells, two separate well plates were prepared to measure (1) the viability at 24 hours via a neutral red reuptake viability assay; and (2) total virus titer at 24 hours using a qPCR against viral DNA.

### 2.8. Neutral Red Reuptake Viability Assay

The neutral red reuptake assay quantitatively measures the number of viable cells. This assay was performed on infected A549, Vero E6, and MCF-7 cells 24 hours post-infection. First, we aspirated the virus-containing media from the wells and washed the cells once with PBS. Subsequently, we diluted 0.3% Neutral Red dye (Sigma-Aldrich, St. Louis, MO, USA) 100-fold in DMEM, added the diluted dye into the wells, and allowed it to incubate in a humidified incubator at 37 °C, 5% CO_2_ for one hour and a half, during which living cells will take in neutral red. After incubation, we removed the neutral red stain from the cells and then washed cells with PBS. To obtain a representative viability, we released the neutral red from live cells by first adding 50 µL of Sorenson’s citrate buffer and then adding 50 µL of 200 proof ethanol. The amount of neutral red dye present is directly correlated with the number of live cells in culture. We measured the amount of neutral red in each well using a BioTek plate reader by measuring the absorbance of the solution in the 540 nm wavelength.

### 2.9. qPCR

Quantitative polymerase chain reaction (qPCR) was used for the quantification of the number of oVVs or EDM-coated oVVs present. Here, in this study, it is used in several instances. (1) To measure the concentration of EDM-coated oVVs in the extrusion product to ensure that the encapsulation did not exclude any oVVs; (2) To measure the total number of copies of oVVs that would be present in A549, Vero E6, and MCF-7 cells 24 hours post infection with their respective experimental sample; (3) Concentration of oVVs or EDM-coated oVVs present in mouse blood, organs, or tumor in their respective in vivo experiments.

We extracted oVV DNA following the manufacturers protocol (Zymo Viral DNA kit). Briefly, we added a volume of vDNA extraction buffer equivalent to 4 times the plated volume to each well of a 24-well plate and incubated for 10 minutes. The samples were transferred to IC column tubes, centrifuged, and washed two times with DNA wash buffer. Then, we eluted the extracted oVV DNA with 20 µL of elution buffer. Once the oVV DNA was extracted from the column tubes, we added 2 uL of the oVV DNA to 18 µL of a SYBR green containing “one step” Mastermix (Applied Biosystems, Waltham, MA, USA) with 1 µmol forward and reverse primer against the Vaccinia E3L gene. The sequences used were: forward-primer 5′-TCCGTCGATGTCTACACAGG-3′ and reverse-primer 5′-ATGTATCCCGCGAAAAATCA-3′ [31]. A standard curve of known amounts of oVV DNA correlated to the infectious units of the input DNA (pfu/mL) was included in each qPCR analysis. For the qPCR, we used an Applied Biosystems Stratagene MX3005P detecting SYBR green signal with a program of 1 cycle of 10 minutes at 95 °C, 40 cycles of 30 seconds at 95 °C, 1 minute at 55 °C, 1 minute at 72 °C.

### 2.10. In Vivo Study: Acute Pharmacokinetics and Biodistribution of oVV and EDM-Coated oVV in Wild-Type Mice upon Intravenous Administration

oVV and EDM-coated oVV inoculums were prepared as described using mouse EDM and injected intravenously into 24 healthy 6-week-old female C57/BL6 mice per experimental group. Six mice from each experimental group were then sacrificed in groups at the following timepoints: 15 minutes, 1 hour, 2 hours, and 4 hours, and the maximum amount of blood was withdrawn via aortic puncture. Whole blood was then used for qPCR in accordance with the protocol outlined in 2.8. All animal experiments were conducted at Anticancer Inc. (San Diego, CA, USA). The sacrificed mice were then dissected, and its various organs (liver, lung, kidney, spleen, and heart) extracted. The organs were then frozen at −80 °C until they were ready for tissue digestion, upon which viral DNA was then quantified.

### 2.11. Tissue Digestion for oVV DNA Extraction

To prepare the mouse tissue for DNA extraction and subsequently qPCR, we first cut up to 100 mg of organ tissue into small pieces, placed organ pieces into a 1.5 mL microfuge tubes, and recorded the weight of each sample. To begin the digestion, the sliced tissue is then submerged in 400 µL a proteinase-k buffer containing 190 µL DNase free water, 190 µL 2X proteinase K buffer, and 20 µL of proteinase K. We then incubated the tissue in a water bath set at 55 °C for 2 hours. Then, we centrifuged the 1.5 microfuge tubes to pellet the tissue debris and extracted the supernatant to collect vDNA for qPCR. We ran qPCR using the same primers, program, and procedure as previously described.

### 2.12. In Vivo Xenograft Study: Surgical Orthotopic Implantation

Tumor xenografts were generated orthotopically using surgical protocols as described elsewhere. In brief, 6 female NOG mice (5–6 weeks old) were injected with a single dose of 5 × 10^6^ MDA-MB-231 human breast cancer cells that have been genetically engineered to stably express red fluorescent protein (RFP) into their second left mammary fat pad and allowed to grow. Once the tumor reached ~50 mm^3^, they were ready for dosing. Mice were sorted into 3 equal groups to receive EDM, oVV, or EDM-coated oVV; the concentration of viral inoculums was equivalent to 1 × 10^7^ pfu in 100 uL PBS of oVV-GFP. Human EDM was used for the EDM-only control, and EDM-coated oVV. Tumors were monitored for size by 3 direction caliper measurements to compute total volume. Fluorescent imaging for green fluorescent protein (GFP) and RFP was captured, and each mouse was monitored for weight for a total of 2 weeks. All animal experiments were conducted at Anticancer Inc. (San Diego, CA, USA). The fluorescent imaging was digitally processed to remove autofluorescent signal from the skin and fur of the mice. A digital binary mask is then generated using signal in the red fluorescent channel gating for presence of cancer cells, and then used to look for co-localizations of GFP.

## 3. Results

### 3.1. Extrusion and Characterization of Coated Virus Product

Since we know that the original oVV contains an uncharacterized mix of enveloped and unenveloped viruses, we can expect that the extruded product will contain oVV, EDM-coated oVV, and empty vesicles. Using a rabbit polyclonal antibody that binds to oVV (both enveloped and unenveloped forms), and the pre-processed oVV/EDM mixture, we were able to generate an ELISA standard curve that shows a linear relationship between the amount of detectable oVV and its resulting colormetric signal (Figure 2A). We then measured the amount of oVV that remains detectable after coating, which then enables us to deduce the amount of undetectable oVV as a result of coating with EDM. Through this, we find that only a small proportion, ~6%, of the oVV remain detectable in the extrusion product, which leads us to believe that more than 94% of the viruses are coated in our process (Figure 2B). Our findings also show a small non-significant increase in the concentration of vaccinia virus in the extrusion product (Figure 2C), suggesting that most of the vaccinia viruses were successfully extruded through the membrane.

To measure the consistency of the extrusion process, we characterized the product size of the extruded product in contrast with the encapsulating membrane material and the starting virus across a triplicate of runs (Figure 2D, Figure 2E, and Figure 2F). It is worth noting that the oVV sample exhibited a bi-modal distribution, one at their native 300–400 nm size, and a smaller broad peak beginning at approximately 1000–2000 nm, which is indicative of aggregation. Due to the presence of two separate peaks, the measured polydispersity index (PDI) is 0.643. The membrane, however, shows a single broad peak at approximately 900 nm, with a PDI of 0.244. The extruded product, however, shows a slightly less uniform distribution at approximately 440 nm, with a PDI of 0.274.

### 3.2. Modeling pH-Based Stimuli Responsiveness in EDM Using Particle Size and Zeta Potential

The presence of sialic acid residues that can exist in an equilibrium comprising protonated and deprotonated residues presents an opportunity to study the attractive and repulsive forces governing EDM stability at different pHs (Figure 3A,B).

As we incrementally change the pH from 7.5 (neutral) to ~4.5 (acidic), we observed that the zeta potential of EDM increase in a nonlinear fashion, which can only be described as gradual between the range of 7.5–6, going from −34 mV to −20 mV, and increasingly steep between 6 and 4.5, going from −20 mV to +5 mV (Figure 4A). Correlating this change in steepness is the observation of a sudden increase in the median particle size of the membranes from their original baseline of 1000 nm to ~2000 nm. This increase then continues to trend upwards to sizes as large as ~4000 nm as the pH continues decreases, signifying aggregation in the nanoparticles (Figure 4B). To verify that this is not a function of concentration, we diluted the EDM solution to show that the trends continue to hold in dilute dispersions. To ensure that we observe the aggregation at its equilibrium state, we make our measurements at varying times after changing the pH. Our results show a very big overlap between the measured sizes at different times suggesting that we are at the equilibrium state.

To better understand the relationship between zeta-potential and particle size, we set up a framework of analysis to quantify a relative repulsive force between the particles. This shows the general trend of diminishing repulsive force with increasing acidity. Our results here very coarsely show an exponential relationship between the relative repulsive force and decreasing pH from 6 to 4. Specifically, within the range of pH 6 to 4, for every decrease in pH of 1, which corresponds to an increase in H+ concentration of one order of magnitude, we observe a decrease in the relative repulsive force of two orders of magnitude. It is also at this pH range that the repulsive forces have undergone a decrease in an order of magnitude relative to the physiological pH, suggesting that the original repulsive force is an order of magnitude higher than the attractive Van der Waals force at physiological pH (Figure 4C and Figure 4D).

### 3.3. Increased Infectivity of EDM-Coated GFP-oVV in Acidic Conditions In Vitro

Findings of aggregation and fusion of EDM in acidic conditions point to a possibility for a lysosomal-like infection mechanism of the coated virus into cancer cells. To probe this hypothesis, we assayed for cell viability and total virus titer. Cell viability, as assayed using the neutral red reuptake assay, trended downwards across the experimental samples ranging from (1) neutral pH + oVV; (2) acidic pH + oVV; (3) neutral pH + EDM-coated oVV; and (4) acidic pH + EDM-coated oVV. For comparison, we provided a negative control for the cells that were unperturbed by any environmental cues, as well as a vehicle + environmental control where we perturbed the system with EDM and acid without introducing any oVV (3). Our results here show much higher viabilities for our controls than for our experimental samples. For the oVV, our results only show a significant increase in cancer cell killing in the MCF-7 cell line with increasing acidity. However, the most substantial viability decrease emerges from the comparison between the cell killing of the bare virus and coated virus, at both the neutral pH and the acidic pH. This shows that the coating of the virus improves on its ability to kill cancer cells at all pHs. Lastly, by comparing the cell viabilities between neutral and acidic samples of the coated virus, we see a slight but not statistically significant improvement in the killing rate of the cancer cell (Figure 5A, Figure 5B, and Figure 5C).

To better understand the role of environmental cues in viral-mediated oncolysis, we quantified the viral titer in the cell lysate using qPCR to obtain an intracellular viral concentration. The exact concentrations were obtained by fitting the CT values onto a standard curve with known viral concentrations. This viral concentration is then normalized to the neutral red assay readout to normalize our results to the number of remaining cells left in the culture. Here, we see the intracellular virus titers increase across the experimental samples ranging from (1) neutral pH + bare virus; (2) acidic pH + bare virus; (3) neutral pH + coated virus; and (4) acidic pH + coated virus (Figure 5D, Figure 5E, and Figure 5F). We also measured virus titers in our controls as comparison to show the baseline noise in the absence of any viruses. (3) We consistently registered only a slight increase in virus concentration between groups 1 and 2, where the differences lie only in the acidity of the environment. Upon coating, the virus titer increases by several folds, suggesting that coating the viruses led to an immediate increase in infectivity, thus leading to an increase in the number of viruses produced. However, we see a further increase in the virus concentration between the sample that was incubated with coated viruses at neutral pH in comparison to the sample that was incubated with coated viruses at an acidic pH, showing that the acidic environment further improved on the viruses’ infectivity. This corroborates our model and our cell viability assay results. In summary, our qPCR results show an almost negative and directly proportional relationship between the cell viability and virus titer, suggesting that cell death is causatively related to the number of viruses that were produced over the same period.

### 3.4. EDM-Coated oVV Evades Circulatory Clearance in Intravenous Administrations

Past studies have shown that cell membrane-coated nanoparticles resist clearance and were able to persist in circulation for longer without sequestration into off-target organs [29]. Here, we observed that the EDM-coated oVV exhibited the same trend when injected into immune competent wild-type mice. We see that both oVV and EDM-coated oVV undergo an initial drop in concentration between 15 minutes and 1 hour. The viral concentration then stops and levels off for the remainder of the time, suggesting that the remaining oVVs/EDM-coated oVVs resist clearance. Overall, EDM-coated oVVs were circulating at a concentration that is almost an order of magnitude higher than that of the unprocessed oVVs (Figure 6A).

To allay concerns that the coating of the virus alters its physicochemical properties, and thus diverts the viral payload to off-target organs where it becomes ineffective, we characterize the biodistribution of the virus across various organs of the mouse at the terminal timepoint of our PK study. Our results here show that both oVV and EDM-coated oVV share similar biodistributions, except for the liver, where EDM-coated oVV is present in lower concentrations (Figure 6B). To compensate for the lack of accumulation in the liver, we see the virus present in higher concentrations in blood. We also observed no viruses, EDM-coated or not, accumulating in the kidney at the time of sampling. This suggests that by coating our viruses, the coated viruses have resisted accumulation in the liver and have preferentially partitioned to the circulatory system.

### 3.5. Increased Oncolytic Activity of EDM-Coated GFP-oVV in Tumor Xenograft Mouse Model

Given the results from our in vitro studies, we then sought to understand its physiological correlation as a treatment in an in vivo cancer model. Here, we chose the MDA-MB-231 breast cancer tumor as it has been shown to establish acidic tumors, and its stable expression of RFP will allow us to monitor the progression of the cancer in real time without harming the mouse. The viruses are injected intratumorally in accordance with our timeline (Figure 7A). This places the virus directly into the target, circumventing the differences between the tumor accumulation of the bare and coated oncolytic virus.

In tracking the size and intensity RFP signal, imaging results at day 7 qualitatively show that while the virus by itself can suppress the growth of the tumor; however, the coated virus is able to exert a greater influence on the tumor and can almost eliminate it (Figure 7B, left). This result is corroborated as we track the tumor size. However, we were only able see trace amounts of GFP signal in the tumors that received oVV or EDM-coated oVV, with no qualitative difference between the two.

In our external tumor size measurements, we also observed at 7 days post-injection that the tumors that received the uncoated virus continue to measure similarly to the negative control in size, while the tumors that were injected with the coated viruses had begun to diverge and shrink in size. This increment in tumor shrinking rate indicates that the coated viruses were able to infect the cancer cells much faster and initiate their replication before the uncoated viruses were able to do so. The uncoated viruses were also eventually able to suppress the growth of the tumor to equal the oncolysis of the coated virus but taking seven more days to do so. This suggests that it took a longer time for the number of uncoated viruses to build up to a threshold concentration for the oncolysis to take place (Figure 7C).

## 4. Discussion

Our work here builds on previous studies showing that the encapsulation of oncolytic viruses with cell membranes improves the infectivity of oncolytic viruses both in vitro and in vivo [26,27]. Past studies have shown that cancer cell-derived membranes can be used to coat oncolytic adenoviruses, creating a non-viral enveloped adenovirus, that exhibits improved cancer cell killing capabilities that look to surpass that of even viral protein-mediated infection through viral envelopes. This has led us to believe that artificial, non-viral envelopes may possess more than one avenue of infection, and that these routes can work synergistically to facilitate viral entry to cells more quickly compared to native viral envelopes. Here, we highlight the possibility that one such avenue could be electrochemical, mediated through pH stimuli, which allows the virus to behave like a “smart” nanoparticle.

To generate these non-viral enveloped OVs, we used erythrocyte membrane as the non-virus envelope. Erythrocyte membrane derivation has been fine-tuned over decades since it was discovered and is well characterized. Erythrocyte membranes also bear many hallmarks of its original cell, including the expression of cell surface markers that allow it to remain in systemic circulation for longer periods of time. This could be important for the systemic administration of OVs.

Our studies show that, after encapsulation, the EDM undergoes consistent reduction in size from 1000 nm to a very uniform distribution of a new EDM-coated oVV with an even distribution peaking at a size of 450 nm, with a low PDI of 0.274 (Figure 2F). It is also worth noting that a particle size measurement of the oVV produced multi-modal distribution with larger particles peaking between 1000 nm and 2000 nm, suggesting that our starting oVV material, while purified of cellular debris in the sucrose gradient, is prone to aggregation (Figure 2E). Coating oVV with EDM produced a much more homogenous product that is stable and resists aggregation.

It is important to note here that at the last step of oVV production, lysis of the cells used to grow the oVVs, releases both IMVs and EEVs and the all the oVVs are collected after purification. Studies have shown that these IMVs often comprise the majority of the entire viral population, with EEVs constituting the remainder [32]. The composition of the product is often not characterized, as separating the IMVs from EEVs is difficult to accomplish on a commercial scale and may not be economically feasible given the larger proportion of IMVs in the product. Here, we conservatively choose not to make any assumptions about the envelope state of the oVV, and as such, for our ELISA, we chose a polyclonal antibody which would in theory bind to both the IMV and EEV in our oVV. We also clearly see that the oVV signal diminishes significantly after coating, which shows that extrusion with EDM has led to lower detectable proportions of both IMV and EEV in the product. Hence, we conclude that we were able to perform the coating of our oVV consistently with >94% efficiency (Figure 2B).

By coating the oVV with EDM, we recapitulate the functionality granted to the virus by the envelope. The envelope is a key component for many enveloped viruses as viral surface proteins directs infection. While EDM-coating does not bear any viral proteins, it can disguise the virus as an erythrocyte, which is nonimmunogenic and, through the expression of CD47 expressed on its surface (otherwise known as the “don’t eat me” signal), prevents the virus from phagocytosis [24,25]. We further discover that the sialic acid groups that decorate the EDM surface can lead to the particle becoming a pH-sensitive vehicle that can respond to changes in environmental pH (Figure 3A, Figure 3B, Figure 4A and Figure 4B). To understand this pH sensitivity and its influence on the delivery of the EDM-coated oVV into the cancer cell, we built a mathematical model correlating differences in physical EDM behavior to its environmental pH. As the environmental pH decreased, these erythrocyte-derived membranes underwent neutralization in surface charge (Figure 4A), suggesting that the sialic residues that decorate the EDM surface are being protonated in the increasingly acidic environment as illustrated in Figure 3A and Figure 3B. The correlating particle size measurements showed no change as the acidity increased, up to a pH of 6, below which the particles sizes started increasing with decreasing pH (Figure 4B). Based on DLVO theory, we deduce that the repulsive force generated by the remaining negative charges has weakened to a state at which the particles can no longer repel each other [33]. Conversely, the inherent Van der Waals forces between the particles become dominant at this pH, allowing the particles to become more physically drawn to each other. This leads to the aggregation and eventual precipitation of the membrane. This phenomenon is also observed in liposomal literature, where dispersion stability is also breached between ±20 mV, below which liposomes also tend to crash out of solution [34,35,36]. To characterize these interactions more quantitatively, we developed a model that considers the surface charge and interparticle distance to estimate and study the trends in repulsive force across the pH of interest. As the interparticle distance starts increasing due to aggregation at a pH of 6, we see that repulsive force between particles start to drop much faster due to the inverse relationship between force and distance in the Coulombic equation (Figure 4C and Figure 4D). In healthy, cellular states, this proposed phenomenon cannot be realized to the dominant repulsive electrostatic forces that push cell membranes apart. However, this is not the same in the acidic tumor microenvironment. We believe that this aggregation of particles leads to an alternative mode of infection via non-endogenous membrane fusion, resembling that of the liposomal delivery of coated payloads. This allows the coated-oVV to be released into the cell.

To test our hypothesis in a controlled environment, we treated several cell lines in vitro following the previously defined experimental groups of: (1) neutral pH + bare virus; (2) acidic pH + bare virus; (3) neutral pH + coated virus; and (4) acidic pH + coated virus. We also prepared negative controls with no treatment and environmental/vehicle controls where the cells are subject to pH perturbations in the presence of EDM but without any viruses. No significant differences were observed between the negative control and the environmental/vehicle control, which shows that neither the pH nor the presence of EDM were cytotoxic to the cells. Minimal differences were observed between experimental groups 1 and 2, with group 2 presenting as slightly more cytotoxic (Figure 5A, Figure 5B, and Figure 5C). This suggests that the virus is not inherently pH sensitive. However, it is worth noting that past studies have shown that A25 and A26, which are mature vaccinia virion proteins that mediate its uptake into a host, are pH sensitive and perform better in acidic environments [37,38]. Since our results show that oVV performs equally in both groups regardless of acidity, this is suggestive that a bigger proportion of the oVV in our oVV samples consists of IMVs.

While our results also yielded no significant differences in viability between the coated virus groups, we detected significant differences in the intracellular viral concentration whether the infection was achieved under acidic or neutral conditions (Figure 5A–F). To quantify intracellular viral concentration, we not only extrapolated the amount of PFUs of oVV within the cells in culture, we also normalized it to the neutral red signal, which we use here to determine the number of living cells in the culture. We felt that this makes the results more meaningful in this context due to the rapid killing capabilities of oVVs in in vitro culture; the more effective the virus became (as a result of the coating), the fewer the number of cells remained in culture, which then prevented the replication of more viruses. In our experimental observations, the viability in both coated virus-treated samples were extremely poor regardless of whether it belonged to the acidic group or the neutral group; most of the well had copious amounts of cell death in the center of the well, leaving only a ring of live cells around the edges. This abundance in cell killing led to our difficulty in statistically distinguishing between the coated samples. However, despite the equivalence in cell death, our intracellular viral concentration assay yielded a significant difference, wherein the cells that were exposed to the coated virus under the acidic condition presented higher viral titers (Figure 5D, Figure 5E, and Figure 5F). For this, we hypothesize that either or both of the following must occur under acidic conditions: (1) increased viral entry per cell; or (2) increased infectivity rate of the viruses into the cells such that viral replication occurs at an earlier time. Since an increase in infectivity of the oVV was not significant across the different pH conditions, we can conclude that the pH sensitivity is a manifestation of the coating and not an inherent property of the virus.

In the other arm of our study, we continue to explore the retention of circulating viruses that can be conveyed onto the oVV through encapsulation. Manufactured oVV can be cleared or deactivated very quickly by many different mechanisms, such as sequestration into the liver or neutralization by complement proteins [39]. Our experiment comparing the circulation of EDM-coated oVV against oVV shows greater short-term persistence when the coating is applied in immune-competent wild-type mice. We chose wild-type mice here as we wanted to study circulatory retention in an immune competent model while remaining outside of the influence of sequestration within tumors. As such, we specially derived EDM from mice as a material for coating the oVVs to maintain the same physiologic relevance.

Given that the pharmacokinetics of the virus was altered by the coating (Figure 6A), it is then just as important here to then understand that the biodistribution of the EDM-coated oVV could also differ. First, we hypothesized that, due to the erythrocyte coating, we will see an increased accumulation of the virus in the liver and spleen after coating (Figure 6B). We base this hypothesis on the natural degradation sites of erythrocytes, where they can either be broken down in the spleen or the liver. However, our results here also show that apart from a lowered liver accumulation, the virus does not tend to sequester in any other organ that it would not have without the coating. Hence, not only do we see that the coating does not alter the biodistribution in any concerning way, but the converse effect of liver evasion also occurred. We also noted that even though the spleen is a major site of erythrocyte sequestration, this accumulation occurs primarily due to the aging of erythrocytes that leads to immunogenic conformational change in the band-3 protein on the surface of erythrocytes, which is tied to its immune-related clearance. Since there is no increased sequestration of EDM-coated viruses in the spleen at the time of sampling, we think that our process of EDM derivation and application of the coating onto the oVV does not lead to the immunogenic conformational change of band 3 protein within 4 hours, which then allows the EDM-coated oVV to side-step accumulation in the spleen. Additionally, the loss of viral signal in the liver when we compare the oVV to the EDM-coated oVV suggests that the coated viruses gained the ability to resist the physical filtration within the liver where the uncoated virus would not.

To measure the translation of this increase in infectivity into a physiologically relevant in vivo model, we chose (1) the MDA-MB-231 breast cancer xenograft model; and (2) to administer the virus intratumorally. Breast cancer was chosen as studies have shown that it can develop acidic tumors [40], and the NOG mice was used as it is the most immune-compromised mouse model that is currently being studied. These NOG mice have not only lost their adaptive immunity but also have reduced leukocytes and impaired phagocytosis mechanisms. This model was specifically used for two reasons: (1) to reduce the amount of immune clearance of the oVV; and (2) to allow us to deliver the EDM-coated oVV using human extracted membranes. Studies with blood transfusions into various mouse models have shown that the NOG mice would clear our human erythrocytes very slowly, which is most conducive for our experiment. In order to eliminate the differences between the tropism of oVV and EDM-coated oVV that may arise because of the differences in blood virus concentration, and homing to the tumor, we opted to directly inject the viruses into the tumor. Our PCR data from Figure 2C show that both samples contain approximately the same number of viruses by genome copy, so we can be sure that tumors across groups receive the same viral dose. This allows us to remove other influences and study, in isolation, the differences in infectivity of oVV and EDM-coated oVV.

Our fluorescent imaging showed a decrease in RFP signal as we compare between the mice that only received EDM, the mice that received the oVV, and mice that received the EDM-coated oVV. This proves that the EDM-coated virus can trigger cancer cell death faster than the oVV itself. Both groups did not show much GFP, which is a direct result of the rapid cancer cell death upon oVV infection and its subsequent apoptosis. This would cause the transcribed GFP to leak out of the tumor, and it would then cease to be detected (Figure 7B).

The tumors that received EDM-coated oVV demonstrated a much faster tumor shrinking response when compared to the tumors that received the oVV (Figure 7C). This echoes our earlier in vitro conclusion that one or both of two things must have occurred: either the coated viruses were able to enter the cancer cells much faster such that the viral replication process was initiated much earlier, or more copies of the coated virus entered the cancer cells which led to a response of higher magnitude. As we eliminated other confounding in vivo factors that could have led to this difference in tumor response, discussions made from our in vitro experiment are more relevant and similar parallels can be drawn, which is that the biggest contributing factor to increased infectivity is the presence of the EDM coating itself, with further improvements due to acidity.

It is also important here to note that, at the end of the study, both the coated and uncoated viruses managed to achieve the same efficacy, noting that the virus is efficacious, albeit slower to act. Ultimately, our goal here was to demonstrate two things: (1) that biological membrane coatings that are not viral envelope can still assist oVV therapy overcome certain barriers, in this case achieving prolonged circulatory retention without sacrificing its infectivity, and (2) that these coating may present stimuli responsive properties that scientists and clinicians can take advantage of when considering the development of a therapeutic. We also would like to add that, since we utilized a xenograft mouse model, there would be a lack of clearance and neutralization of the native oVV. However, had this study been performed in an immune competent mouse, the differences could have been much bigger because the slower acting therapeutic would be more liable to immune clearance, even when injected intratumorally.

## 5. Conclusions

In conclusion, our results show that coating oVVs with EDM is a promising strategy for addressing oVVs’ clinical limitations. Coated oVVs demonstrated not only the ability to remain in circulation for longer, but also showed enhanced infectivity through either better or faster uptake of the virus into the cells. This improved infectivity has also demonstrated the potential to be enhanced within an acidic environment, which may be a way to target the infection to within the tumor as it can enter cancer cells faster. Our model built on DLVO theory not only suggests that this rapid infection can be further improved through a reduction in external repulsive forces, but also by increasing the attractive forces that the payload has for its target.

## Figures and Tables

**Figure 1 pharmaceutics-14-01810-f001:**
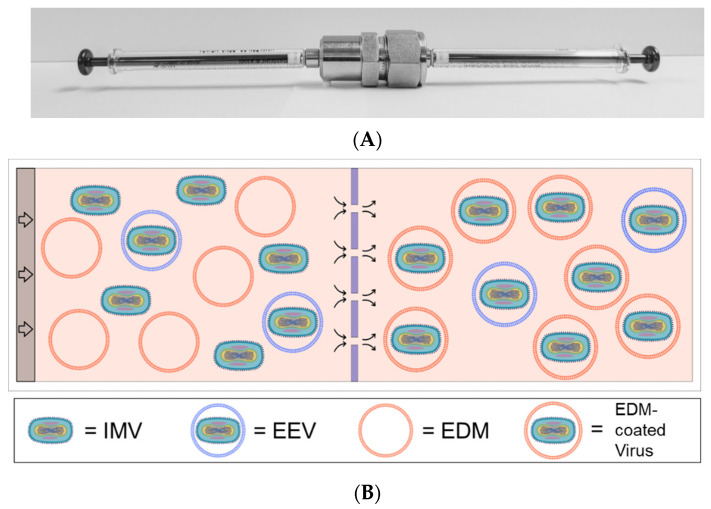
(**A**) Picture of an assembled extrusion system involving an inlet syringe, a membrane holder, and a product syringe. (**B**) Schematic of the extrusion process: oVV, which contains a mix of the unenveloped intracellular mature vaccinia (IMV) and extracellular enveloped vaccinia (EEV), and EDM is loaded on the left and, through the extrusion, emerged as EDM-coated oVV or EEV on the right. (**C**) Depiction of the EDM coating, notably with sialic acid residues decorating its surface.

**Figure 2 pharmaceutics-14-01810-f002:**
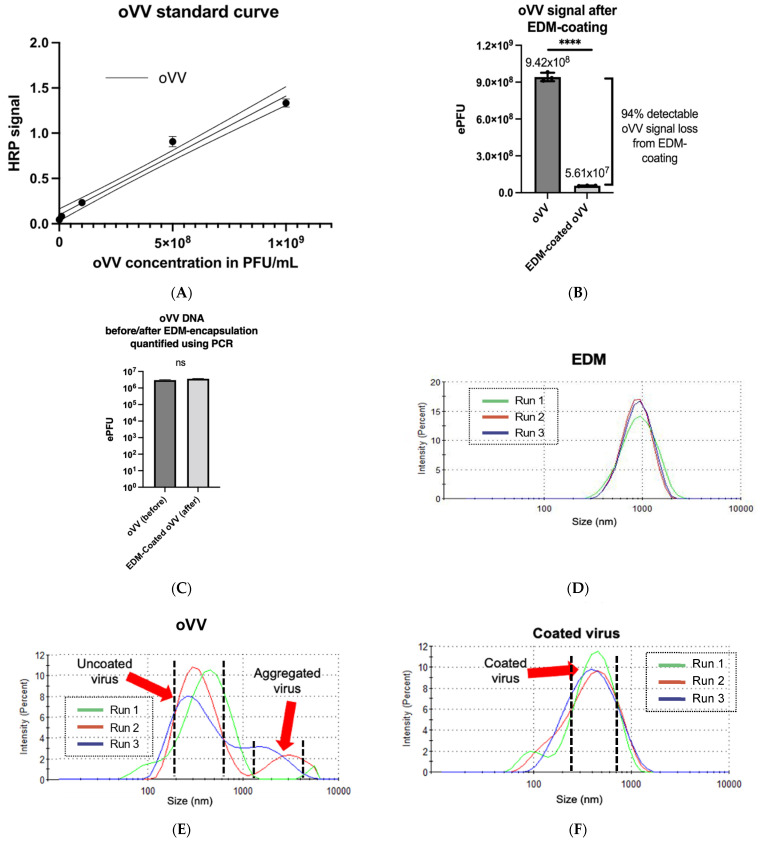
(**A**) oVV standard curve generated via ELISA assay. Dotted lines above and below the solid line denote a 95% confidence in the values of the assay. (**B**) Detectable oVV signal before and after processing with EDM, interpolated from the ELISA standard curve. Results are presented as mean ± SD, *n* = 3. Data were analyzed with ANOVA with Tukey pairwise comparison test. The level of statistical significance was set at probability level **** *p* < 0.0001. (**C**) oVV DNA before and after extrusion as quantified using PCR. Results are presented as mean ± SD, *n* = 3. Data were analyzed with ANOVA with Tukey pairwise comparison test. The level of statistical significance was set at probability level ns *p* ≥ 0.05. (**D**–**F**) Particle size measurements of the (**D**) EDM, (**E**) oVV, and (**F**) EDM-coated oVV, measured across 3 dynamic light scattering runs.

**Figure 3 pharmaceutics-14-01810-f003:**
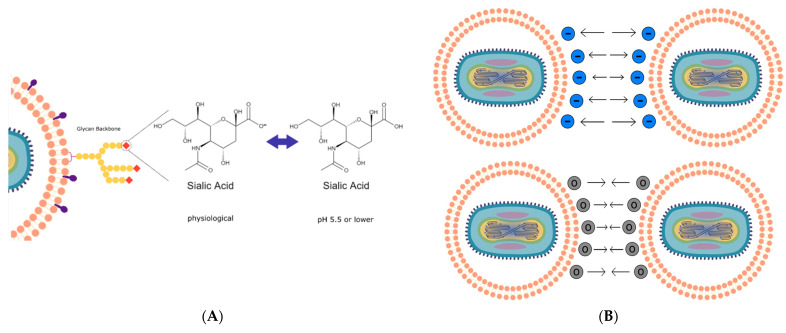
(**A**) Schematic of the basis of our model: protonation of sialic acid residues at any pH lower than the physiological pH neutralizes the strongly negative charges. (**B**) Diagram showing lowering the electrostatic repulsion between the particles via charge neutralization. DLVO theory predicts a threshold where the attractive Van der Waals forces will bring particles together in aggregation.

**Figure 4 pharmaceutics-14-01810-f004:**
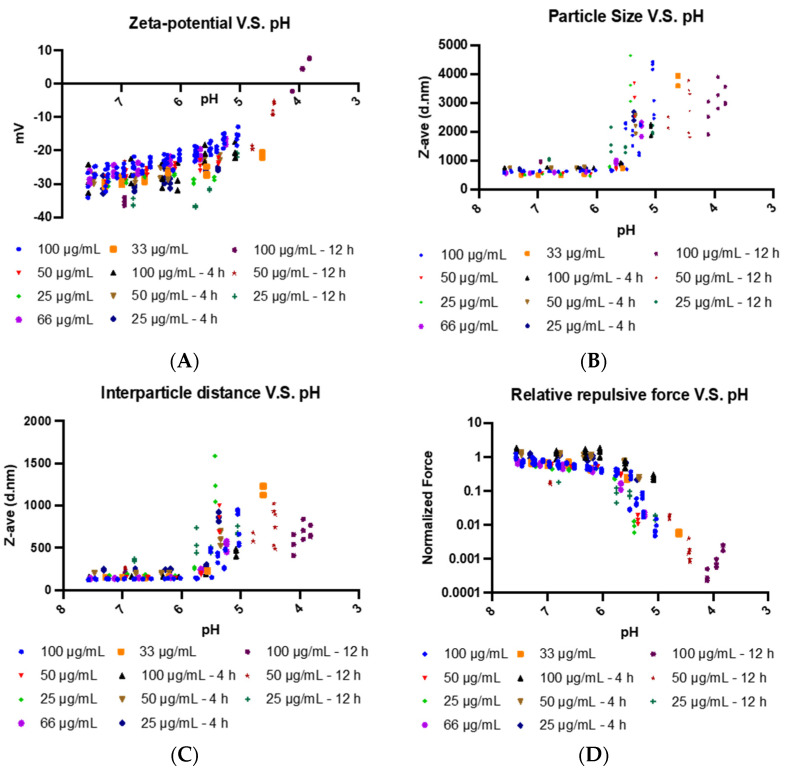
As the pH decreases and the environment becomes more acidic: (**A**) zeta-potential increases; (**B**) particle size increases at pH < 6; (**C**) interparticle distance increases at pH < 6; and (**D**) relative repulsive forces slowly decrease at pH < 6 and decrease much faster at pH < 5.

**Figure 5 pharmaceutics-14-01810-f005:**
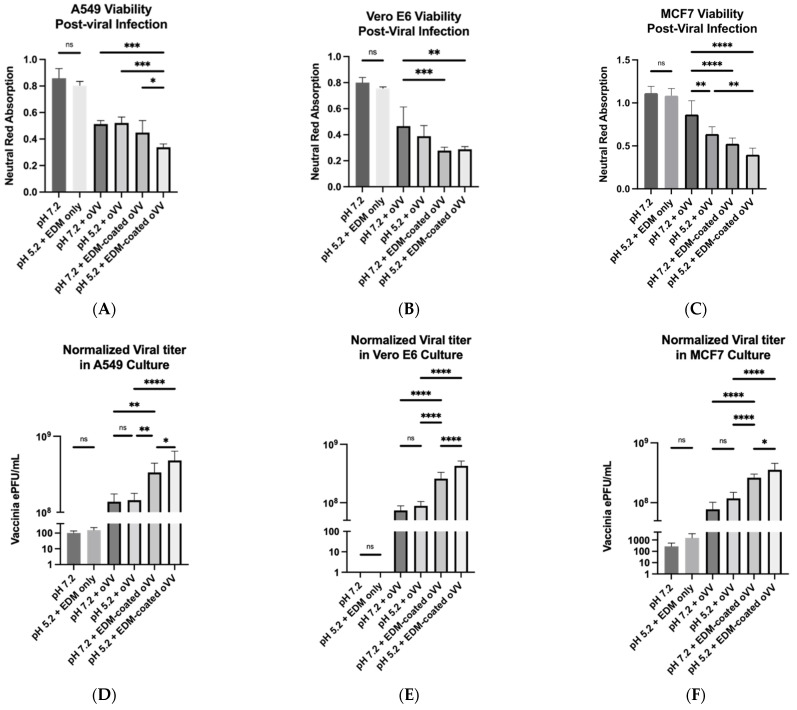
(**A**–**C**) Neutral red-based viability assay results of various cancer cell lines when oVV/EDM-coated oVV were used to infect at different acidities, along with their accompanying negative controls and vehicle/pH controls; (**D**–**F**) intracellular viral concentrations at 24 hours post-infection of various cancer cell lines by oVV/EDM-coated oVV at different acidities along with their accompanying vehicle/negative controls. All results are presented as mean ± SD, *n* = 6. Data were analyzed with ANOVA with Tukey pairwise comparison test. The level of statistical significance was set at probability level ns *p* ≥ 0.05, * *p* < 0.05, ** *p* < 0.01, *** *p* < 0.001, **** *p* < 0.0001.

**Figure 6 pharmaceutics-14-01810-f006:**
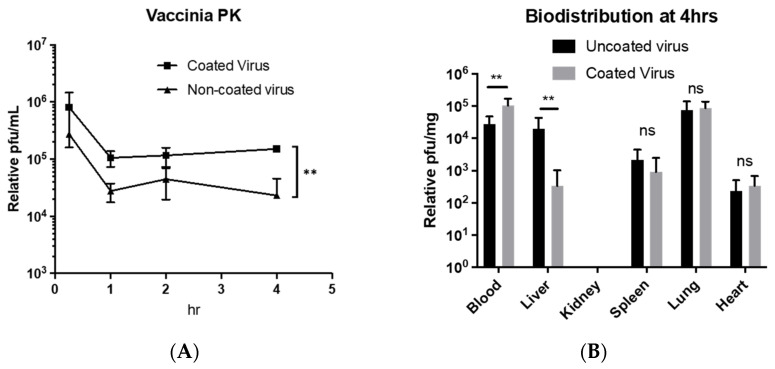
(**A**) Acute pharmacokinetics of an intravenously administered dose of coated and uncoated oncolytic vaccinia virus. (**B**) Biodistribution of the coated and uncoated oncolytic vaccinia viruses in blood and in several organs 4 hours post-intravenous administration. The results are presented as mean ± SD, *n* = 6. Data were analyzed with Student’s *t*-test. The level of statistical significance was set at probability level ns *p* ≥ 0.05, ** *p* < 0.01.

**Figure 7 pharmaceutics-14-01810-f007:**
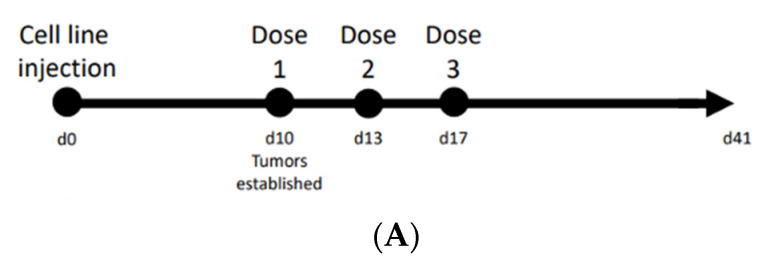
(**A**) Timeline of the comparison study between coated and uncoated oncolytic vaccinia study in an MDA-MB-231 xenograft model. (**B**) (**left**) Fluorescent image of MDA-MB-231 stably expressing RFP marks the tumor in live mice, while (**middle**) eGFP denotes viral protein expression in the tumor. (**right**) Merged images of RFP and GFP. Scale bar denotes 5 mm. (**C**) Tumor sizes monitored over 2 weeks after the first dose was administered showed more rapid responses from the coated oVV at week 1, followed by the subsequent clearing of the tumor at week 2. The results are presented as mean ± SD, *n* = 5. Data were analyzed with ANOVA with Tukey pairwise comparison test. The level of statistical significance was set at probability level * *p* < 0.05, ** *p* < 0.01.

## Data Availability

Not applicable.

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
