# Peer review of "pH Sensitive Erythrocyte-Derived Membrane for Acute Systemic Retention and Increased Infectivity of Coated Oncolytic Vaccinia Virus"

_pharmaceutics, 2022, doi:10.3390/pharmaceutics14091810_

Round 1

Reviewer 1 Report

The manuscript “pH-responsive Erythrocyte-Derived Membrane for Immune 2 Evasion and Rapid Infection of Coated Oncolytic 3 Vaccinia Virus” describes a new strategy for oVV envelope engineering. The manuscript is overall well written and introduces an interesting technology for oVV encapsulations, with far reaching implications for viral tropism engineering and cancer therapy. The manuscript describes oVV encapsulation into erythrocyte-derived membranes (EDM), aided by modelling and in vitro testing of pH-dependent effects on transduction and oncolytic efficiencies. In vivo experiments have also been performed to highlight the oncolytic properties of EDM-oVV compared to their bare-oVV counterpart although EDM enveloping seems to speed-up the tumour shrinking process, it does not lead to a significant therapeutical benefit over bare-viruses on the longer term, suggesting that the effects of EDM-pseudotyping are short-lived.

While the pH-dependent infection rates are well supported by modelling and in vitro testing, there is not enough in vivo evidence to claim that EDM-enveloped oVV display an increased homing in tumour tissues due to the experimental setup (intra-tumor injection). Additionally, while the immune evasion is plausible (given the EDM-envelope) there is not enough evidence for an increased persistence of EDM-oVV in vivo after systemic injection, but only evidence for an increased homing in the blood compartment, possibly mediated by the presence of the envelope and not necessarily by an EDM-one. The claim of immune evasion should therefore be removed from the title. Additionally, human derived EDM are used for in vivo mouse experiments, with the speculation that they should help shield the virus from the immune system although it is fair to suppose that the mouse immune system would readily recognize and attack the “non-self” proteins present on the human derived oVV coating.

Although the manuscript presents an interesting strategy, certain claims (some of which highlighted in the title) are, in my view, not fully supported by the data. Additionally, figures are not always clear and easy to understand. In some instances, important controls should be added where possible to exclude secondary effects of EDM only treatment.

For instance, an EDM only control (without oVV) should be included in the cell viability assay to exclude EDM-mediated toxicity on target cells, mediating the increase in cell death.

Throughout the manuscript, a recombinant oVV expressing eGFP is used. This valuable tool however has not been used extensively. Monitoring eGFP expression could have provided valuable insight into infection dynamics and replication of EDV-oVV over bare-oVV, helping to elucidate if the observed differences are due to enhanced infection rates. While the authors correctly discuss that envelopes mediate better infections, they did not thoroughly investigate this in the manuscript. The pH sensitivity of EDV could indeed explain things, but the sole presence of an envelope, regardless of if EDV-related or not, could indeed explain most of the findings described in the manuscript.

Overall, the manuscript provides evidence (although weak) for a beneficial effect of EDM-enveloping in enhancing infection and oncolytic rates both in cell culture and, to a lesser extent, improve tumor-shrinking kinetics in vivo. I am however not fully convinced by the quality of the data and, although I recognise the novelty of the work, I have to admit there is not enough evidence that strongly support an improvement of oVV therapy in presence of an EDM-encapsulation.

Specific comments (methodologies)

1.       How was the oVV obtained and purified? Please include amplification and purification methodologies in a separate “Methods” section. Also, the oVV appears to be recombinant, as there is mention of “eGFP”. Please include, if available, a relevant citation of the full sequence of the recombinant oVV in Supplementary data.

2.       As per my (very limited) knowledge of oVV, and as also suggested repeatedly by the schematics included in this manuscript, VACV-oVV are enveloped DNA virus. If only bare-viruses were used for encapsulation experiments, the drawings must be changed accordingly. Inclusion of oVV purification methodologies is extremely important for downstream implications. Indeed if oVV are a mixture of both bare and enveloped viruses, EDM enveloped oVV with mixed membrane properties can arise after encapsulation. If the ELISA abs are specific for bare viruses, both wt enveloped-BV and the EMD-oVV could only detected by this method post-permeabilization. A clarification in oVV extraction and purification procedures and ELISA kit specifications is indeed required, including abs specificity for enveloped and non-enveloped oVVs.

3.       The authors mention and ELISA assay performed manufacturer’s recommendations. Manufacturer or catalogue number for either antibodies or kit are however not specified. It is not clear if the antibodies used in this kit do recognise bare- or enveloped-oVV.

4.       Encapsulation procedures – The schematics provided consistently depicts enveloped oVVs. Specification about bared/enveloped oVV composition prior encapsulation is required (please see previous points). If enveloped viruses are present prior encapsulation, how did the author prevent membrane shearing and wt-envelope/EDM mixing during encapsulation? If enveloped wt-oVV are used, or not excluded, the pictures in Fig1B-C are likely to be misleading, as the chance of having EDM encircling an intact enveloped viruses are dim, and a mixed envelope is more likely resulting from the shearing process.

Figures-related comments

Figure 2A – the graph is not clear and it appears that not all the data are depicted. How many replicates per data point? What is the histogram bar representing and why it is depicted differently from the rest of the data? Methods mention standard curve with both bare and enveloped oVV, why these data are not depicted? Discussion mentions that EDM-oVV could not be detected by ELISA, due to the envelope shielding the abs from binding to the capsid. Where are these data represented? It is not clear from this graph.

Figure 2B – Legend, what is the quantification method used? Plaque assay? How many replicates?

Figure 2C/D/E – Color-codes are not explained in Figures (using a legend) or in Figure legend. Arrows are indicating intersection points which could be attributed to any of the three data points. Not clear.  Are these replicates?

Figure 2D – how the authors can be sure the “aggregated virus” is truly aggregate or indeed wild-type enveloped viruses?

Figure 4-B/C/D – Wrong Y axis labels, not matching graphs titles. Additionally, it is not clear from text if these measurements were performed on EDM-alone or EDM-oVV after encapsulation. If EDM-alone were used, where this processed through extrusion as described for other experiments? Please include more details about these experiments in either text or Figure Legend.

Figure 5D-F – I am not sure the normalization procedure used for plotting these histograms. qPCR data must not be normalised for the neutral red absorption. They can either be plotted as absolute ePFU/ml (without normalisation) or, if the author wish, an housekeeping gene can be used and data could be presented as viral copies per genome. If the viral DNA extraction procedure used does not allow to isolate genomic DNA, a mitochondrial amplicon can be used instead. However, shown like this, the results are in my opinion not reliable and potentially misleading.

Figure 6A – Are the authors sure that the titers of both coated and non-coated viruses were properly measured at time point 0? The reason I’m asking is that, not accounting for T0, both coated and uncoated virus display a very similar PK trend. Differences could indeed be due to artifacts in viral titration. How can the author exclude this?

Figure 6B – are any of these results significant? Significance levels have not been included in this plot. Additionally, chart title says Biodistribution @1 hr. “@” should be “at”. Also figure legend says that Figure 6B depicts biodistribution at 4 hrs. Which one is it?

Figure 7B – The Virus encodes eGFP. What is then the “vehicle” control? The name suggests a “no virus condition” but can the author explain what is the eGFP expression attributable to? Also eGFP and RFP should be presented separately and not only as merge. A scalebar is also missing. Also two of the panels have been used for commercial purposes ahead of print on the company which founded this work. I am not sure this is standard practice and I leave to the editor the comment on this matter.

Reviewer 2 Report

The manuscript  by Kaelan T. Samoranos et al. deals with a promising approach in antitumor therapy  based on the oncolytic properties of certain human viruses (e.g.  Vaccinia Virus). Authors are presenting convincing data concerning incorporating of Vaccinia Virus into erythrocyte derived nanoparticles which possibly could protect virus from host immunity, lengthen its half-life and, therefore potentiate its antitumor activity. Experimental design is clear and appropriate. The conclusions made by the authors logically follow from the results obtained. The manuscript may certainly be of interest to the readers of Pharmaceutics. However some minor point might be addressed to improve the article:

1 In the Figure 2 C-E, it is not clear exactly which different size distributions are indicated by the green, blue, and red lines.

2 Despite kidney is mentioned in Figure 6B, there are no appropriate data for virus distribution.

3 Since the spleen is the major site of erythrocyte sequestration one can expect that erythrocyte membrane particles should have increased uptake by spleen. How do authors explain low targeting of coated virus to spleen in their experiments? 

Author Response

To our esteemed reviewer,

Our team at Coastar would like to thank you for your review and comments. We have revised our manuscript in order to rectify some of the items you have pointed out.

Reviewer: In the Figure 2 C-E, it is not clear exactly which different size distributions are indicated by the green, blue, and red lines.

Response: We have since revised Figure 2 with color-coded legends.

Reviewer: Despite Kidney is mentioned in Figure 6B, there are no appropriate data for virus distribution.

Response: We did not manage to detect any viral DNA (via qPCR) from the extract of the kidneys. As such we believe that there is no sequestration of virus in the kidney.

Reviewer: Since the spleen is the major site of erythrocyte sequestration one can expect that erythrocyte membrane particles should have increased uptake by spleen. How do authors explain low targeting of coated virus to spleen in their experiments?

Response: We thank you for pointing this out. Previous studies have shown that accumulation of erythrocytes within the spleen is due to immune-linked clearance of erythrocytes following antibody binding to Band-3 protein that has undergone confirmational change. This process of confirmational change takes place over a span of time that exceeds the timeframe of our pharmacokinetics study (4 hours post-injection).

Again, we at Coastar thank you for your time and for reviewing our manuscript.

Best,

Coastar Therapeutics

Round 2

Reviewer 1 Report

Response to the authors

I appreciate the changes that the authors included in the current version of the manuscript. I with to thank the authors for the time and levels of detail with which they addressed my comments. I overall think that the manuscript has enormously benefited from this round of revision. Although there are few points which I would still like to be addressed, these concern mostly explanation and rephrasing, and a couple of clarifications about some of the new figure panels. I would recommend this work for publication following a (very) minor revision.

 Note: My point by point follows, I have kept all past comments, and included new ones in Italic as indented paragraphs for clarity. Many of the points were exhaustively addressed, and therefore there’s not comment follow-up on the, although rest assured, I am completely satisfied by that specific answer.

 Additional note for (potential) future iterations: If another round of revision is requested by the editor, I would require, for the next round of revision, a "clean" pdf of the final manuscript too as it's quite difficult to follow line numbering and figure formatting on a track-change only manuscript. 

 ------------------------------------------------------------------------------

To our esteemed reviewer,

The team at Coastar would like to extend our thanks for your thorough reading and constructive criticism of our manuscript. We have revised the manuscript as directed and feel that our quality of work has improved because of the feedback and are hence very grateful.

Before diving into a point-by-point reflection of our amendments to the manuscript, we would like to just highlight here that our motivation here is to point out that erythrocyte membrane coating, which has been applied to nanoparticles in the past to keep them in circulation and prevent them from clearance, can also be used to keep viruses in circulation. While the encapsulation of cell membranes onto viruses is not novel, as it has been shown to work with Adenoviruses in cancer cell membranes in the past, our combination of Vaccinia virus and Erythrocyte membrane is, which, as we seek to show, improves circulatory retention of the virus in a way that a cancer cell membrane coated virus would not.  More importantly, we would like to bring to attention the pH-responsiveness of the membrane particles themselves.

On a scientific note, we felt that this area was particularly interesting, and a relatively untouched area in viral formulations. By putting this out there, and as you have alluded to yourself, we would like to suggest that improving viral tropism for cancers need not be limited to modifications to the virus but can be approached more wholistically from multiple avenues. As such, we have changed our title to more clearly reflect our intentions and to be more in-line with the multi-functional theme of the special issue, which is that such coatings can both prevent acute systemic clearance and confer pH responsivity to the virus.

Reviewer: The manuscript “pH-responsive Erythrocyte-Derived Membrane for Immune Evasion and Rapid Infection of Coated Oncolytic  Vaccinia Virus” describes a new strategy for oVV envelope engineering. The manuscript is overall well written and introduces an interesting technology for oVV encapsulations, with far reaching implications for viral tropism engineering and cancer therapy. The manuscript describes oVV encapsulation into erythrocyte-derived membranes (EDM), aided by modelling and in vitro testing of pH-dependent effects on transduction and oncolytic efficiencies. In vivo experiments have also been performed to highlight the oncolytic properties of EDM-oVV compared to their bare-oVV counterpart although EDM enveloping seems to speed-up the tumour shrinking process, it does not lead to a significant therapeutical benefit over bare-viruses on the longer term, suggesting that the effects of EDM-pseudotyping are short-lived.

While the pH-dependent infection rates are well supported by modelling and in vitro testing, there is not enough in vivo evidence to claim that EDM-enveloped oVV display an increased homing in tumour tissues due to the experimental setup (intra-tumor injection).

Response: We’d like to thank reviewer for the comment. As pointed out by the reviewer, here we have selected to use an immune deficient xenograft model, and have chosen to administer our therapeutic intra-tumorally. For this study, we are trying to demonstrate the improved internalization instead of homing of the EDM-coated oVV into the tumor. We felt that one of the obvious criticisms of EDM-coating onto the virus is the blocking or elimination of viral protein mediated infection, and our in vivo model here aims to address that instead of preventing viral infection, we are seeing improved infection, which in this case, we propose is mediated by the EDM-coating itself. In parallel hypotheticals, had we chosen to administer this via IV and observed a reduction in tumor, then the conclusion there would have been, as the reviewer suggested, due to increased homing, which is beyond the scope of this study. We have amended our discussion in response to this point so that our intent is more clearly conveyed. This change can be found in lines 592-600 and 614-624.

Reviewer: I understand the point raised by the authors and their answer. The difference between homing and internalization can be subtle to unspecialized readers. I do agree with the authors if “internalization” translates to “the ability of the virus to penetrate the cells” or “infection efficiency” which is more broadly used. The use of “infection efficiency” instead of internalization might help clarify this point, as internalization might be broadly confused with homing in the context of whole organisms (e.g. internalization within a tissue, or in this case a tumour).

Reviewer: Additionally, while the immune evasion is plausible (given the EDM-envelope) there is not enough evidence for an increased persistence of EDM-oVV in vivo after systemic injection, but only evidence for an increased homing in the blood compartment, possibly mediated by the presence of the envelope and not necessarily by an EDM-one. The claim of immune evasion should therefore be removed from the title.

Response: We agree with the reviewer’s point and have changed our title to reflect this.

Reviewer: Additionally, human derived EDM are used for in vivo mouse experiments, with the speculation that they should help shield the virus from the immune system although it is fair to suppose that the mouse immune system would readily recognize and attack the “non-self” proteins present on the human derived oVV coating.

Response: We apologize for the miscommunication. Human-derived EDM was used only in the immune deficient xenograft model. Mouse-derived EDM was used for our pharmacokinetic and biodistribution study in the wild-type mice to more properly reflect a physiological condition. The choice the use Human-derived EDM in the xenograft was made based on a couple factors: (1) the tumor grew out from human cell lines, and (2) studies of human blood clearance in mice that showed that human blood is actually cleared sufficiently slowly in nude mice. We postulated that if we wanted to study infectivity into human cancer cells by oVVs under more relevant conditions, we would be better served using EDM derived from humans. To make this more clear, we have made the necessary amendments in our paper. This change is reflected in sections 2.2, 2.10 and 2.12.

Reviewer: I did not recall this detail in material and methods of the first draft of the manuscript but I might have misread. This clearly is clearly more biologically relevant and I am not questioning the use of human EDM in the context of injection in tumors of human origin, although grafted in mice. If mouse EDM have been used for systemic EDM-oVV injection in mice the experiments have no flaws from this aspect. Thanks for the clarification.

Reviewer: Although the manuscript presents an interesting strategy, certain claims (some of which highlighted in the title) are, in my view, not fully supported by the data. Additionally, figures are not always clear and easy to understand. In some instances, important controls should be added where possible to exclude secondary effects of EDM only treatment.

For instance, an EDM only control (without oVV) should be included in the cell viability assay to exclude EDM-mediated toxicity on target cells, mediating the increase in cell death.

Response: We appreciate the reviewer’s comment. We’ve added on additional negative vehicle and conditional controls to show the individual effects of our temporal pH shock, and then the combined effect of pH + EDM, in addition to the proposed negative control of EDM-only. The authors once again thank the reviewer for suggesting this. Figure 5 has been updated accordingly with its accompanying results section.

Reviewer: Apologies ahead if I am not correctly referring to the right panel/figure. Due to the track changes present in my version of the manuscript is impossible to follow on lines indications and figures might be affected too. Despite this, I appreciate the inclusion of EDM controls only and while there is reassuringly no effect on Neutral Red absorption, I am puzzled to see that Vaccinia virus can be detected in these conditions by PCR (histograms showing vaccinia pfu/ml) in A549 and MCF7 cells, while Vero cells are clean. Can the authors comment on this?

Reviewer: Throughout the manuscript, a recombinant oVV expressing eGFP is used. This valuable tool however has not been used extensively. Monitoring eGFP expression could have provided valuable insight into infection dynamics and replication of EDV-oVV over bare-oVV, helping to elucidate if the observed differences are due to enhanced infection rates.

Response: We appreciate reviewer’s insight. We have tried to track eGFP expression without much success. Our experience with this oVV is that the oncolysis occurs at a very rapid rate with the cells in the wells. The neutral red cell viability data presented here is obtained from cells at a time point of 16-18 hours post-infection, and as you can see, most of the cells are already dead at this point, with most of the remaining cells exhibiting fluorescence, irregardless of pH or coating. Without access to an incubated, live-monitoring microscope, it was difficult for us to track the emergence of GFP signal in real time. When we moved the experiment in vivo, we tried to capture the eGFP expression within the tumors; we present this data in our discussion (lines 603-608)

Reviewer: I appreciate the technical challenges of working with an oncolytic virus and the lack of access to an incubated microscope for real-time eGFP tracking. There are other ways to monitor eGFP expression (e.g. western blot or qRT-PCR) but they would be undoubtedly more tedious and, in the context of the manuscript, would not add a lot. I was more curious about why the eGFP monitoring was not used although present in the viral vector, and the authors have provided an exhaustive response.

Reviewer: While the authors correctly discuss that envelopes mediate better infections, they did not thoroughly investigate this in the manuscript. The pH sensitivity of EDV could indeed explain things, but the sole presence of an envelope, regardless of if EDV-related or not, could indeed explain most of the findings described in the manuscript.

Response:

Currently, the vaccinia viral envelope is thought to mediate better infections through viral protein A27, which binds onto cell surface heparan sulfate to facilitate its uptake. This brings about two independent issues, the first of which is that these oncolytic vaccinia viruses are typically generated in Vero E6, A549, or HeLa cells, which means that the A27 containing viral envelope will also carry cancer cell markers that are immunogenic and will cause the enveloped virus to be rapidly cleared. The second issue, which is perhaps the most contentious one, is that the bulk of the infectious viruses, during the large scale manufacturing process are unenveloped vaccinia viruses, and that the composition of enveloped and unenveloped virus amongst the viral manufacturing product is not typically characterized, as the separation of the two can be challenging given the current state of technology (this separation can be achieved more easily and accurately with lab scale viral preps). Also, since the bulk of the infectious viruses are unenveloped viruses, even if the technology to separate the viruses exist, selectively separating only enveloped viruses would mean discarding most of the product, which on a commercial scale, makes the therapeutic unfeasible economically.

Recent studies have shown that cell membranes derived from other sources, such as cancer cells, can be used to coat adenoviruses to artificially create an “enveloped adenovirus”. These studies go on to show that these non-viral enveloped adenoviruses exhibit higher infectivity and better oncolysis, and have outlined the mechanisms for that in their studies. As this has already been studied, here in this paper, we cite their studies to partially explain why our EDM-coated viruses (hereafter referred to as EDM-coated oVV) outperform the oVV that we’ve received from the manufacturers, rather than re-probing the mechanism. (4) we have rephrased parts of our manuscript to reflect this more clearly. To this end, the reviewer is completely correct, and that having an envelope is better than not having an envelope, be it artificially coated or not. However, as artificial non-viral envelopes do not bear viral proteins that typically mediate viral infection, we feel that there might be other modes of infection, asides from the ones that have been shown in literature, that can work synergistically to improve infectivity of non-viral envelopes above that of the viral envelopes. This line of questioning has then led us to our discovery that environmental pH can serve as a cue to improve infection of viruses with non-viral envelopes.

To summarize, a non-viral envelope, such as the one we have demonstrated here, can serve several purposes. (1) To a mixed and uncharacterized population of oVVs, coating them all will make them more homogenous by coating the unenveloped oVVs artificially. (2) Improve on their ability to stay in circulation if they were to be administered IV has non-viral envelopes will neither contain viral proteins nor cancer cell membrane proteins. (3) grant it multiple, synergistic routes of infecting their target cell, of which one of them is the ability to be triggered by pH-based cues.

Here in this study, we aim to demonstrate (2) improved circulatory retention, and also bring to attention (3) that depending on the choice of the non-viral envelope, we can grant the coated virus the ability to infect on pH stimuli, leaving non-pH-based envelope-mediated infectivity to be discussed elsewhere. We had previously discussed this briefly at the beginning of the discussion section, but we’ve further added to it for clarity. This change is reflected in lines 461-468 and lines 482-493. We also address the enveloped oVV and unenveloped oVV composition proportions in Figure 1.

Reviewer: I thank the authors for the exhaustive and instructive answer. This was previously not entirely understandable from the manuscript alone (for a non-oVV expert audience). Including some of this notions in the discussion section is definitely improving the manuscript and I’m happy about this point. Also the changes made on Figure 1 help to clearly understand the dynamics of the coating process and the different initial and final species potentially present in the mix.

Reviewer: Overall, the manuscript provides evidence (although weak) for a beneficial effect of EDM-enveloping in enhancing infection and oncolytic rates both in cell culture and, to a lesser extent, improve tumor-shrinking kinetics in vivo. I am however not fully convinced by the quality of the data and, although I recognise the novelty of the work, I have to admit there is not enough evidence that strongly support an improvement of oVV therapy in presence of an EDM-encapsulation

Response: We appreciate reviewer’s comment. While we can understand why the reviewer feels that there is no beneficial effect, as both the oVV and EDM-coated oVV both led to the same outcomes, the goal of our study was not to show an immediate beneficial effect of EDM-coating to the extent that it can be used for multiple administrative avenues as a silver bullet against cancer. Here, our goal is simply to demonstrate that membrane coatings can assist oVV therapy overcome certain barriers, in this case achieve prolonged circulatory retention, as stated in our conclusion. We also show that the membrane can grant the oVV certain physical/electrical/chemical properties that scientists and clinicians can take advantage of, in this case pH sensitivity for more selective infectivity into cells in acidic tumor microenvironments. On top of that, we are also showing that these characteristics that we grant to the oVV do not come at a detriment, and that EDM-coated oVV can function and behave in many ways, like the manufactured oVV prior to EDM-encapsulation; they continue to cause tumor shrinkage (albeit at a faster rate), and that they even share similar biodistributions in vivo.

Reviewer: Having more clearly specified the scope of this work, I agree with the authors, and I have no further objections on this. However I encourage the authors to include some of these considerations in discussion if they deem it necessary.

Specific comments (methodologies)

Reviewer: How was the oVV obtained and purified? Please include amplification and purification methodologies in a separate “Methods” section. Also, the oVV appears to be recombinant, as there is mention of “eGFP”. Please include, if available, a relevant citation of the full sequence of the recombinant oVV in Supplementary data.

Response: We appreciate reviewer’s comment. We have added a reference that corresponds more specifically to the manufacturing of the virus, and we have included details as to the amplification and purification of the virus in section 2.1.

Reviewer: As per my (very limited) knowledge of oVV, and as also suggested repeatedly by the schematics included in this manuscript, VACV-oVV are enveloped DNA virus. If only bare-viruses were used for encapsulation experiments, the drawings must be changed accordingly. 

Response: We appreciate reviewer’s comment. This is indeed important, and we have updated figure 1 to more accurately reflect the diverse population of the pre-processed oVV.

Reviewer: Inclusion of oVV purification methodologies is extremely important for downstream implications.

Response: We have added a reference that corresponds more specifically to the manufacturing of the virus, and we have included details as to the amplification and purification of the virus in section 2.1.

Reviewer: Indeed if oVV are a mixture of both bare and enveloped viruses, EDM enveloped oVV with mixed membrane properties can arise after encapsulation.

Response: We appreciate reviewer’s comment. This is correct. However, the EDM:oVV ratio by mass is roughly on the order of 100:1; as we use EDM in excess to ensure that as many of the oVV particles can be coated with EDM as possible. Even as we generate mixed membranes, its likely that the EDM:oVV envelope will overwhelmingly contain EDM. We believe that our conjecture is supported by our ELISA study for reasons listed below. This is discussed in lines 482-493.

Reviewer: If the ELISA abs are specific for bare viruses, both wt enveloped-BV and the EDM-oVV could only detected by this method post-permeabilization. A clarification in oVV extraction and purification procedures and ELISA kit specifications is indeed required, including abs specificity for enveloped and non-enveloped oVVs.

Response: We appreciate reviewer’s comment. The ELISA antibody is the same polyclonal antibody generated against whole virus, which we made the mistake of putting it under our “reagents” section in 2.1 as opposed to where it should have gone in 2.4. We’ve since re-categorized it and expounded on our methods so that the ELISA steps can be more clear. Because the ELISA antibody is a polyclonal generated against the whole virus, we believe that it will bind to both the enveloped-oVV, which will have both viral proteins and cancer-associated proteins on its surface, as well as the un-enveloped oVV, bare capsid protein. That was in fact a huge motivating factor in us choosing this antibody, which is for its widespread capture. If I may point to our revised ELISA results figure, you will see that there is still remnant binding onto the EDM-coated oVV that is approximately 5% that of what the original signal would have been. In this case, one of 2 situations must have taken place: (1) if the oVV used for encapsulation contains mostly unenveloped oVV viruses, than we must have coated them with the EDM such that the polyclonal antibody can no longer bind to 95% of the viruses; (2) if the oVV used for encapsulation contains mostly enveloped oVV viruses, then our mixed membrane product must overwhelmingly comprise EDM, which is not reactive against the polyclonal detector antibody. To ensure that this point can be conveyed more clearly across, we’ve also updated our discussion section to bring attention to this point in lines 482-493.

Reviewer: The authors mention and ELISA assay performed manufacturer’s recommendations. Manufacturer or catalogue number for either antibodies or kit are however not specified. It is not clear if the antibodies used in this kit do recognise bare- or enveloped-oVV.

Response: We appreciate reviewer’s comment. We’ve completely re-written our ELISA assay protocol in section 2.4 and furnished it with details, and we will address the recognition targets of the antibody in the discussion as aforementioned.

Reviewer: Encapsulation procedures – The schematics provided consistently depicts enveloped oVVs. Specification about bared/enveloped oVV composition prior encapsulation is required (please see previous points).

Response: We appreciate reviewer’s comment. We apologise for the lack of clarity, we and we have added a legend to explain ourselves better. Our schematics was depicting non-enveloped oVVs. We have updated figure 1 to reflect our current understanding of the initial composition of oVVs.

Reviewer: If enveloped viruses are present prior encapsulation, how did the author prevent membrane shearing and wt-envelope/EDM mixing during encapsulation?

Response: We appreciate reviewer’s comment. Our particle size analysis shows that the prior encapsulation, the oVV population is mostly within range of 300-400 nm, which is consistent with literature. To prevent excessive shearing of the virus, our porous membrane contains pores of 1000nm in width, which allows for 3 lengths of the virus to pass through side-by-side. As fluid dynamics dictates, we will most likely be undergoing laminar flow, as the pore is very small. In this situation, the virus particle is likely going to flow along the convective current, which is most prominent in the middle of the pore where the fluid velocity is highest. This is also where the shear forces are lowest as shear forces are going to be highest on the edge of the pore due to the no-slip boundary conditions imposed. Together, these are the measure that we’ve taken to prevent membrane shearing.

However, even if we were to account for membrane shearing, it would be likely that it would strip the more fluid viral membrane, and leave the oVV capsid in suspension. Without assuming that this membrane is stripped partially or completely, we will then, as the reviewer postulated, very likely encounter viral envelope/EDM mixing. Since we use a lot more EDM than we use viruses by mass, the resulting mix will be a oVV that is enveloped mostly, if not completely with EDM. We think of this as advantageous to keeping the virus in circulation as the viral envelope is immunogenic and hence cannot remaining in circulation for long, in comparison to the EDM membrane. This is further evidenced by the binding of rabbit anti-oVV polyclonal antibody against the oVV product prior to encapsulation that we show in our ELISA assay.

Reviewer: If enveloped wt-oVV are used, or not excluded, the pictures in Fig1B-C are likely to be misleading, as the chance of having EDM encircling an intact enveloped viruses are dim, and a mixed envelope is more likely resulting from the shearing process.

Response: We appreciate reviewer’s comment. This is entirely correct, and we have rectified figure 1 for clarity.

Figure-related comments

Reviewer: Figure 2A – the graph is not clear and it appears that not all the data are depicted. How many replicates per data point?

Response: We apologize for the lack of standard error bars, and we’ve added them back in and we broke the figure down into two separate figures, 2A and 2B for clarity. To address the reviewer’s question, the standard curve was done in triplicates.

Reviewer: What is the histogram bar representing and why it is depicted differently from the rest of the data?

Response: The histogram was meant to represent the proportion of viruses that did emerged undetected after the EDM-coating process, as a subtracted from the number of viruses that remained detectable after the EDM-coating process. We have revised figure 2A and 2B for clarity.

Reviewer: Methods mention standard curve with both bare and enveloped oVV, why these data are not depicted?

Response: The standard curve is done with the manufactured oVV, which contains a mix of the enveloped and unenveloped virus, and this standard curve is presented in figure 2A.

Reviewer: Discussion mentions that EDM-oVV could not be detected by ELISA, due to the envelope shielding the abs from binding to the capsid. Where are these data represented? It is not clear from this graph.

Response: We have modified the figure and broke it down to show the amount of EDM-oVV that reacted with the antibody in the ELISA in Figure 2B, and correspondingly, also show the flip side, which is the amount of oVV that became obscured from binding because of it has become EDM-coated

Reviewer: Figure 2B – Legend, what is the quantification method used? Plaque assay? How many replicates?

Response: The quantification method used is qPCR against viral DNA, and we added this information into the figure caption. We measure the number of genome copies of oVV before and after our processing method. The PCR was done at n=3 as a technical triplicate.

Reviewer: Color-codes are not explained in Figures (using a legend) or in Figure legend. Arrows are indicating intersection points which could be attributed to any of the three data points. Not clear.  Are these replicates?

Response: We apologise for the poor explanation. The color codes are different runs of the same experimental sample. The arrows actually point to all three runs. We’ve added legends to address this.

Reviewer: Figure 2D – how the authors can be sure the “aggregated virus” is truly aggregate or indeed wild-type enveloped viruses?

Response: Wild-type enveloped viruses have been found in literature to be almost on the same order of size as the un-enveloped virus, which is between 300-400 nm. However, here we see the particles balloon to a size of more than 1000nm, which is too big for it to be an enveloped virus. Additionally, of the many vaccinia viruses that we received, most of them exhibit considerable clumping in the thawed suspension, which goes away after a little bit of mild bath sonication. As such, we believe that this the emergence of the peaks between 1000nm and 2000nm belong to aggregates. Also, given that this particular sample has been purified through a sucrose gradient, it is unlikely for these to be cell debris, which would have been (mostly) removed in that process.

Reviewer: Figure 4-B/C/D – Wrong Y axis labels, not matching graphs titles. Additionally, it is not clear from text if these measurements were performed on EDM-alone or EDM-oVV after encapsulation. If EDM-alone were used, where this processed through extrusion as described for other experiments? Please include more details about these experiments in either text or Figure Legend.

Response: We sincerely apologise for this mistake. We will make the necessary corrections. The measurements were done on EDM-alone, and we’ve furnished the much needed details in our text.

Reviewer: Figure 5D-F – I am not sure the normalization procedure used for plotting these histograms. qPCR data must not be normalised for the neutral red absorption. They can either be plotted as absolute ePFU/ml (without normalisation) or, if the author wish, an housekeeping gene can be used and data could be presented as viral copies per genome. If the viral DNA extraction procedure used does not allow to isolate genomic DNA, a mitochondrial amplicon can be used instead. However, shown like this, the results are in my opinion not reliable and potentially misleading.

Response: We appreciate reviewer’s comment. We would like to advocate for this normalization here, even though it is not commonly done, because the neutral red assay allows us to normalize the viral DNA to the number of living cells remaining in the culture as a result of viral oncolysis. To establish the context, our protocol for analysis is as follows: to the cell culture, we remove the overlaying media (which contains oVV, cell debris and dead cells from oncolysis), rinse it thoroughly with PBS, before extracting the DNA out from the remaining attached cells. However, due to the oncolysis, as you can see from the neutral red study, there are discrepancies in the number of cells remaining in each experimental sample. If we were to simply quantify this with qPCR and report it without accounting for the number of remaining cells, it would not be a good comparison as there are fewer cells the more infectious the virus was because of the experimental treatment.

Reviewer: I totally understand the rationale behind the normalization, I am however not sure this is widely accepted for publication. If normalization by PCR is not possible anymore (e.g. the samples are not available or else), it would probably better not to include these panels and just proceed showing the neutral red staining. Text would have to be changed accordingly but it would be more credible from a scientific soundness perspective. I have very rarely (if not never) seen any normalization methods using two different techniques to plot the data in the same graph. I am just saying this in the interest of the authors and it only represents my opinion. I would let the authors decide on this how to best represent their results.

The reviewer brought up potential housekeeping genes as a possibility, and our viral DNA extraction kit does indeed also extract whole cell genomes. However, we must bear in mind that unlike RT-PCR, we will not be dealing with a steadily expressed protein, like GAPDH, or b-actin, of which we can obtain multiple copies of its transcriptome. This mode of cell number quantification then is rather obscure and there is a general lack of well verified housekeeping genes for this specific application, as technically we can target any region in the genome, for every cell will only have 1 copy of it. We hesistate to take a stab at this considering we will have to re-design primers to include both introns and exons within the genomic code, and instead chose just to normalize to number of remaining cells as shown by neutral red.

Reviewer: This is incorrect, what I am suggesting is not to normalize viral DNA over genomic transcript, but rather using genomic DNA. Exon and intron must not be considered in this case, as it would make no sense. Any genomic region would be equally good as housekeeping gene, as there’s only two copies of any given locus in the cells. There is absolutely nor problem in detecting one copy of a given housekeeping gene. This technique is routinely used, for instance, to quantify by qPCR the number of transgene integrations with transposase or lentivirus, and it would perfectly apply also in this case. This would indeed provide a rather strong normalization approach. The absolute cell number doesn’t need to be determined, and the virus could simply be express as pfu/genome copies. As previously specified, I understand the reasons behind neutral red absorption normalization and I would not argue further about the necessity for a standardized normalization method, although it would be advisable. The authors can choose if keeping this normalization method as is, or even remove the panels altogether as they deem best.

Past studies have also attempted to use the absolute quantity of DNA as a way of determining how many cells are remaining. However, this is also unavailable to us considering that our entire DNA fraction would be contaminated with viral DNA.

Reviewer: This is true, but for instance western blot could be a cheap alternative for relative quantification, as it would provide both information about any given housekeeping protein (e.g. GAPDH, ACTB) and oVV proteins detected through the polyclonal ab already used for ELISA. Relative quantification would be extremely easy in this case. As both protein species can be measured with the same experiments and technique.  

Reviewer: Figure 6A – Are the authors sure that the titers of both coated and non-coated viruses were properly measured at time point 0? The reason I’m asking is that, not accounting for T0, both coated and uncoated virus display a very similar PK trend. Differences could indeed be due to artifacts in viral titration. How can the author exclude this?

Response: Our value for time point 0 is obtained from theoretical calculation, which we estimate by assuming that the concentration within the injection (which we obtain by qPCR) will be diluted by the total volume of mouse blood in the animal, which studies have shown is approximately 2 mLs. It was logistically difficult to obtain mouse blood right after we injected it. Our first real measurement is obtained at t=15 mins. In hindsight, this was not the most accurate thing to do, and as such, we’ve removed the t=0 time point from our figure.

Reviewer: Figure 6B – are any of these results significant? Significance levels have not been included in this plot. Additionally, chart title says Biodistribution @1 hr. “@” should be “at”. Also figure legend says that Figure 6B depicts biodistribution at 4 hrs. Which one is it?

Response: Sorry for the title. It is 4 hours and we’ve made the necessary corrections and added in our statistical analyses. The first 2 groups for Blood and Liver are statistically significant (**, p<0.01)

Reviewer: Figure 7B – The Virus encodes eGFP. What is then the “vehicle” control? The name suggests a “no virus condition” but can the author explain what is the eGFP expression attributable to? Also eGFP and RFP should be presented separately and not only as merge. A scalebar is also missing. Also two of the panels have been used for commercial purposes ahead of print on the company which founded this work. I am not sure this is standard practice and I leave to the editor the comment on this matter.

Response: The Vehicle control is an injection of EDM in PBS. We re-did this figure to focus only on the tumor and presented the two color layers separately, as well as add the scalebar to the image.

Reviewer: Thanks for specifying the nature of the control. Can the author explain why the control was expressing eGFP in the previous version of the manuscript, given than it should contain no virus?

Reviewer: Also two of the panels have been used for commercial purposes ahead of print on the company which founded this work. I am not sure this is standard practice and I leave to the editor the comment on this matter.

Response: Since we have changed the figure in Figure 7, they are now different figures and there should no longer be a conflict in the publishing rights to the figures.

Reviewer: Thanks to the authors for updating the figures so that there’s no additional overlap with panels previously published for commercial purposes.
